# A Deep Architecture for Log-Linear Models

**Simon Luo, Sally Cripps**
School of Mathematics and Statistics
The University of Sydney
Data Analytics for Resources and Environments (DARE)
Australian Research Council
Sydney, Australia
s.luo@sydney.edu.au,sally.cripps@sydney.edu.au

**Mahito Sugiyama**
National Institute of Informatics
JST, PRESTO
Tokyo, Japan
mahito@nii.ac.jp

## Abstract

We present a novel perspective on deep learning architectures using a partial order structure, which is naturally incorporated into the information geometric formulation of the log-linear model. Our formulation provides a different perspective of deep learning by realizing the bias and weights as different layers on our partial order structure. This formulation of the neural network does not require any gradients and can efficiently estimate the parameters using the EM algorithm.

## 1 Introduction

Log-linear models and energy based models (EBMs) are a family of models which are widely used in statistics and machine learning. Log-linear models provide the relationship between probabilistic models and structured frameworks, such as a graphical model. In this paper, we investigate the use of log-linear models as a means of framing a deep-learning architecture in a probabilistic manner. Our approach is based on the $em$-projection algorithm (which stands for exponential-mixture) to estimate parameters of the log-linear model (Amari, 1995). The $m$-projection estimates the expectation of the latent variables given the current parameters of the model, while the $e$-projection estimates the parameters in the model via maximum likelihood estimation (MLE). (Amari, 1995), show that the $em$-projection algorithm in information geometry coincides with the statistical EM algorithm (which stands for Expectation-Maximization) (Dempster et al., 1977), in a majority of cases.

However, the $em$-projection algorithm usually requires an iterative approach to estimate the parameters in the $e$-projection (Amari, 1995), making the algorithm less efficient than the iterative approach. In this paper, we formulate the deep-learning architecture as a poset (partially ordered set) (Gierz et al., 2003). The partial order structure decomposes the model-parameters into two hierarchical layers that represent the bias and the edge weights to allow for a closed form analytical expression for the for the maximum likelihood estimation which is typically computed by the $e$-projection in the projection algorithm or M-step in the statistical EM algorithm. Our proposed approach allows us to efficiently update parameters without any gradients and therefore does not have any issues such as vanishing gradients are not encountered in our formulation.

## 2 Preliminary on Neural Networks

A perceptron (Rosenblatt, 1958) is a mathematical neuron defined as an element which receives $M^{(t-1)} \in \mathbb{Z}^+$ inputs $\mathbf{u}^{(t-1)} = [u_1^{(t-1)}, \ldots, u_{M^{(t-1)}}^{(t-1)}] \in \mathbb{R}^{M^{(t-1)}}$ and emits an output $o_j^{(t)} \in \mathbb{R}$, where $t$ denotes the layer of the current neuron and $t-1$ denotes the layer with all the incoming neurons, and $M^{(t-1)}$ is the number of neurons at layer $t-1$. Parameters $w_i^{(t-1)} \in \mathbb{R}$ of the neuron represents the edge weights of each input $u_i^{(t-1)} \in \mathbb{R}$ and $b_j^{(t)} \in \mathbb{R}$ to be the threshold for activation.

Then the integrated input of neuron $j$ at layer $t$, $u_j^{(t)}$ is calculated by the weighted sum of inputs at layer $t-1$ and is given by

$$u_j^{(t)} = \sum_{i=1}^{M^{(t-1)}} w_{ij}^{(t-1)} o_i^{(t-1)} - b_j^{(t)} = \sum_{i=1}^{M^{(t-1)}} w_{ij}^{(t-1)} \sigma \left( u_i^{(t-1)} \right) - b_j^{(t)}, \tag{1}$$

where the output of the neuron, given some activation function $\sigma$, is computed by $o_j^{(t)} = \sigma \left( u_j^{(t)} \right)$, where $\sigma$ is typically chosen to be a sigmoid function or a Rectified linear Unit (ReLU) for the hidden layers and a softmax function for the output layer. Perceptrons can be hierarchically stacked with each other to construct a multi-layered perception where each layer is denoted as $t$ with a total of $T$ layers in the entire network. This definition of the multi-layered perception is often referred to as a vanilla neural network.

For this study, we define the input at layer $t = 0$ to be $\mathbf{x} = [x_1, \ldots, x_{M^{(0)}}] \in \mathbb{R}^{M^{(0)}}$, where $x_k = o_k^{(0)} = \sigma(u_k^{(0)})$ with the activation function that linearly scales the feature between 0 and 1. We denote the output of the hidden layers to be $\mathbf{h}^{(t)} = [h_1^{(t)}, \ldots, h_{M^{(t)}}^{(t)}] \in \mathbb{R}^{M^{(t)}}$ for $t \in \{1, \ldots, T-1\}$, where $h_k^{(t)} = o_k^{(t)} = \sigma(u_k^{(t)})$ with the sigmoid or the ReLU activation function for $k = 1, \ldots, M^{(t)}$, for $t = 1, \ldots T-1$. The output for layer $T$ is defined to be $\mathbf{y} = [y_1, \ldots, y_{M^{(T)}}]$, where $y_k = o_k^{(T)} = \sigma(u_k^{(T)})$ with a softmax activation function for classification with the domain $(0, 1)$ and ReLU for regressions with the domain $\mathbb{R}$.

## 3    Connection Between Binary Log-Linear Model and Partial Orders

In this section we introduce the relationship between the log-linear model, (Agresti, 2012), and a poset $(\Omega, \preceq)$ (Sugiyama et al., 2016, 2017). We first let $\Omega$ be the sample space of distributions and $\preceq$ denotes the partial order between elements in $\Omega$. If interactions between variables are represented as a partial order between elements in $\Omega$; that is, if each element in $\Omega$ represents a variable configuration and the resulting set $\Omega$ is a poset, the interactions can naturally be treated as a log-linear model. In the poset we denote the least element by $\perp$ and $\Omega^+ = \Omega \setminus \{\perp\}$. For a given subset, $\mathcal{S} \subseteq \Omega^+$, the log-linear model with parameters $(\theta(s))_{s \in \mathcal{S}}$ is given as,

$$\log p\left(\omega'; \theta\right) = \sum_{\omega \in \Omega^+} \mathbf{1}_{[\omega \preceq \omega']} \theta\left(\omega\right) - \psi(\theta) \tag{2}$$

for each $\omega' \in \Omega$. The parameters $\theta$ in the model coincide with the definition of the natural parameter in the exponential family distribution. The partition function $\psi(\theta) \in \mathbb{R}$ can be uniquely obtained by $\psi(\theta) = \log \sum_{\omega' \in \Omega^+} \exp[\sum_{\omega \in \Omega^+} \mathbf{1}_{[\omega \preceq \omega']} \theta(\omega)] = -\theta(\perp)$. The information geometric structure of the set of distributions, $\mathcal{G} = \{p \mid 0 < p(\omega) < 1 \text{ for all } \omega \in \Omega \text{ and } \sum_{\omega \in \Omega^+} p(\omega) = 1\}$, arises when we introduce the expectation parameter, given as

$$\frac{\partial}{\partial \theta(\omega)} \psi(\theta) = \eta(\omega) = \sum_{\omega' \in \Omega} \mathbf{1}_{[\omega \preceq \omega']} p(\omega'; \theta). \tag{3}$$

Then the pair $(\theta, \eta)$ becomes a dual coordinate system of the statistical manifold $\mathcal{G}$, and they are orthogonal, because they are connected via a Legendre transformation. The orthogonality will be used in optimization of the model, where $\theta$ and $\eta$ are jointly used to achieve minimization of the KL divergence via the projection algorithm.

## 4    Implementing Deep Architecture as Partial Orders

We introduce our key technical contribution by formulating a neural network on a partial order structure. First we define the sample space $\Omega$ of the log-linear model to be $\Omega^+ = \mathcal{X} \cup \mathcal{Y} \cup \mathcal{H} \cup \mathcal{W}$, where $\mathcal{X}$ is the set of input nodes, $\mathcal{Y}$ the set of output nodes, $\mathcal{H}$ the set of hidden nodes and $\mathcal{W}$ the set of edge weights. Each entry in the set is denoted as, $\underline{x} \in \mathcal{X}$, $\underline{y} \in \mathcal{Y}$, $\underline{h} \in \mathcal{H}$ and $\underline{w} \in \mathcal{W}$ representing an input node, an output node, a hidden node, and an edge weight, respectively. We use an underline to distinguish an element of $\Omega$ and the corresponding node in a neural network. We also denote by

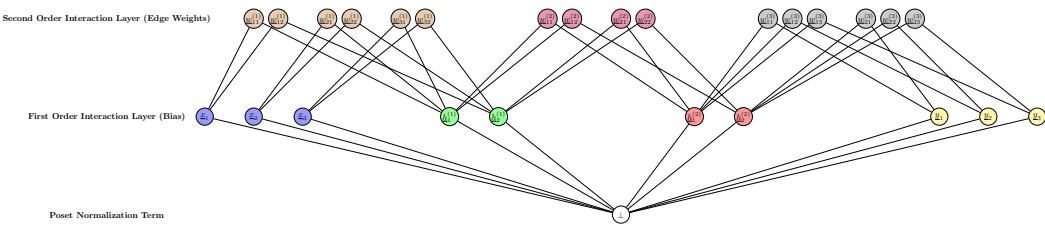

Figure 1: A poset representation of a deep architecture with 3 nodes in the input layer and output layer and 2 hidden layers each with 2 hidden nodes.

$\underline{n} \in \mathcal{N} = \mathcal{X} \cup \mathcal{Y} \cup \mathcal{H}$ the set of visible and hidden nodes in our model. We define the parameters in our model that we estimate to be $\theta$ associated with $s \in \mathcal{S} = \mathcal{H} \cup \mathcal{W}$.

We define a partial order on $\Omega$ inspired by a multi-layered perceptron studied in deep learning to be,

$$
\begin{cases} \underline{n}_i^{(t-1)} \preceq \underline{w}_{ij}^{(t)}, & \text{for all } t \in [0,\, T] \\ \underline{n}_j^{(t)} \preceq \underline{w}_{ij}^{(t)}, & \text{for all } t \in [0,\, T] \\ \underline{n}_k^{(t)} \not\preceq \underline{w}_{ij}^{(t')}, & \text{otherwise.} \end{cases} \quad , \qquad \begin{cases} n_i^t \not\preceq n_j^{t'}, \\ w_{ij}^t \not\preceq n_{kl}^{t'}. \end{cases} \quad . \tag{4}
$$

where the index for each layer is $t \in T$ and $T$ is the total number of layers in the network. The input layer is defined to be $t = 0$ and the output layer is defined to be $t = T$. This partial order structure of $\Omega$ is illustrated in Figure 1 for the neural network architecture shown in Figure 2. We focus on the conventional neural network which has the parameters of the bias and edge weights to represent the first- and second-order interaction, respectively.

We rearrange Equation (1) to define the integrated input $u_k^{(t)}$ of each node $k$ for $k = 1, \ldots, M^{(t)}$ to be

$$
u_k^{(t)} = \epsilon\left(\underline{n}_k; \theta\right) = \log p\left(\underline{n}_k; \theta\right) + \psi\left(\theta\right) = \sum_{s \in \mathcal{S}} \mathbf{1}_{[s \preceq \underline{n}_k]} \theta\left(s\right) = \theta\left(\underline{n}_k\right), \tag{5}
$$

where $\epsilon(\omega; \theta) := \log p(\omega; \theta) + \psi(\theta) = \sum_{s \in \mathcal{S}} \mathbf{1}_{[s \preceq \omega]} \theta(s)$. The value of the edge weight, $w_{ij}^{(t)}$ is

$$
w_{ij}^{(t)} = \epsilon\left(\underline{w}_{ij}^{(t)}; \theta\right) = \sum_{s \in \mathcal{S}} \mathbf{1}_{[s \preceq \underline{w}_{ij}^{(t)}]} \theta\left(s\right) = \theta\left(\underline{n}_i^{(t-1)}\right) + \theta\left(\underline{w}_{ij}^{(t)}\right) + \theta\left(\underline{n}_j^{(t)}\right). \tag{6}
$$

The relationship between the classical neural network and our partially ordered structure is apparent by defining the expectation of each node in $\mathcal{X}$ and $\mathcal{Y}$ to be,

$$
\hat{\eta}\left(\underline{x}_k\right) = \frac{\exp\left(\mathbb{E}\left[x_k\right]\right)}{\sum_i \exp\left(\mathbb{E}\left[x_i\right]\right)}, \qquad \hat{\eta}\left(\underline{y}_k\right) = \frac{\exp\left(\mathbb{E}\left[y_k\right]\right)}{\sum_i \exp\left(\mathbb{E}\left[y_i\right]\right)}. \tag{7}
$$

In the following section we define the optimal values of $\theta$ and $\eta$, and show how to estimate them.

## 5   Optimization on Posets

We optimize the log-linear model by minimizing the KL divergence between the distribution of a given input and output pair $(\mathbf{x}, \mathbf{y}) = \{(x_i, y_i)\}_{i=1}^N$, and the joint distribution $\hat{p}(\mathcal{X}, \mathcal{Y}, \mathcal{H}, \mathcal{W}; \theta)$. The objective function is given as

$$
D_{\text{KL}}\left(p\left(\mathbf{x}, \mathbf{y}\right) \| \hat{p}\left(\mathcal{X}, \mathcal{Y}, \mathcal{H}, \mathcal{W}; \theta\right)\right) = -\mathbb{E}_{p(\mathbf{x},\mathbf{y})}\left[\log \hat{p}\left(\mathcal{H}, \mathcal{W} | \mathcal{X}, \mathcal{Y}; \theta\right)\right] + D_{\text{KL}}\left(p\left(\mathbf{x}, \mathbf{y}\right) \| \hat{p}\left(\mathcal{X}, \mathcal{Y}\right)\right), \tag{8}
$$

see Appendix for the derivation. In Equation 8, $\hat{p}$ is the estimated probability of the poset structure and can be determined from the data points using Equation (7). We note that $p(\mathbf{x}, \mathbf{y}) = \eta(\mathbf{x}, \mathbf{y})$ because there are no hidden elements in our input dataset. Our optimization only requires the first term because the second term does not depend on $\theta$. So our objective function can be written as a function of $\boldsymbol{\theta} = (\theta(s))_{s \in \mathcal{S}}$ and $\boldsymbol{\eta} = (\eta(s))_{s \in \mathcal{S}}$.

$$
J\left(\boldsymbol{\theta}, \boldsymbol{\eta}\right) = -\mathbb{E}_{p(\mathbf{x},\mathbf{y})}\left[\log \hat{p}\left(\mathcal{H}, \mathcal{W} | \mathcal{X}, \mathcal{Y}; \boldsymbol{\theta}\right)\right], \tag{9}
$$

which is equivalent to Maximum Likelihood Estimation (MLE). Our optimization is performed using the statistical EM-algorithm (which stands for expectation-maximization), which alternately iterates M-step and the E-step.

The E-step minimizes the objective function with respect to the expectation parameter $\eta(s)$ and the M-step minimizes the objective function with respect to the natural parameter $\theta(s)$,

$$\boldsymbol{\eta}_{\text{next}} = \arg\min_{\boldsymbol{\eta}} J(\boldsymbol{\theta}, \boldsymbol{\eta}), \qquad \boldsymbol{\theta}_{\text{next}} = \arg\min_{\boldsymbol{\theta}} J(\boldsymbol{\theta}, \boldsymbol{\eta}). \tag{10}$$

We now present the equations for the E-step and the M-step.

## 5.1 Expectation Step

The E-step estimates the expectation of each of the variables, given the parameters estimated at the previous step. The probability estimated by the E-step is, $\hat{p}(\omega; \hat{\theta}) = (1/\exp[\psi(\hat{\boldsymbol{\theta}})]) \exp[\sum_{s \in \mathcal{S}} \mathbf{1}_{[s \preceq \omega]} \hat{\theta}(s)]$.

The expectation of the edge weights can be computed by $\hat{\eta}(\underline{w}_{ij}) = \hat{p}(\underline{w}_{ij}; \hat{\theta}) = (1/\exp[\psi(\hat{\boldsymbol{\theta}})]) \exp[\epsilon(\underline{w}_{ij}; \hat{\theta})]$. We now compute the expectation parameter $\hat{\eta}$ for the input nodes, hidden nodes and the output node

### 5.1.1 Updating Probabilities in the Input and Output Nodes.

The probabilities $\underline{x}_k \in \mathcal{X}$ and $\underline{y}_k \in \mathcal{Y}$ are required to be updated after each step using the logistic function (see Appendix for derivation).

$$\hat{p}(\underline{x}_k) = \frac{\hat{\eta}(\underline{x}_k)}{\left[1 + \sum_j \exp\left[\hat{\theta}(\underline{w}_{kj}) + \hat{\theta}(\underline{h}_j)\right]\right]}, \quad \hat{p}(\underline{y}_k) = \frac{\hat{\eta}(\underline{y}_k)}{\left[1 + \sum_j \exp\left[\hat{\theta}(\underline{w}_{jk}) + \hat{\theta}(\underline{h}_j)\right]\right]}. \tag{11}$$

### 5.1.2 Forward Propagation on a Poset

Forward propagation can be computed directly on the poset structure. From Equation (1), we substitute the parameter values for the edge weights and the neuron to obtain the forward equation to compute the energy of the hidden nodes. This is given by,

$$u_k^{(t)} = \sum_i \left[\left(\hat{\theta}\left(\underline{n}_i^{(t-1)}\right) + \hat{\theta}\left(\underline{w}_{ik}^{(t-1)}\right) + \hat{\theta}\left(\underline{n}_k^{(t)}\right)\right)\right] \sigma\left[\hat{\theta}\left(\underline{n}_i^{(t-1)}\right)\right] = \hat{\theta}\left(\underline{n}_k^{(t)}\right). \tag{12}$$

See Appendix for details. Then the expectation of the hidden node can be computed as

$$\hat{\eta}\left(\underline{h}_k^{(t)}\right) = \sum_i \hat{p}\left(\underline{w}_{ik}^{(t-1)}; \hat{\theta}\right) + \sum_j \hat{p}\left(\underline{w}_{kj}^{(t)}; \hat{\theta}\right) + \hat{p}\left(\underline{h}_k^{(t)}; \hat{\theta}\right). \tag{13}$$

The forward propagation is used to estimate the parameters in the model during the training phase. It is an optimizer for Equation (10) because it updates $\boldsymbol{\eta}$ given the model parameters $\theta$ in the previous $e$-projection. After the model has been trained, the forward propagation is also used to make predictions by propagating the values of the input through the network until it reaches the output layer.

## 5.2 Maximization Step

The M-step computes the MLE of the parameters given the expectation of the variables in the model. For the second-order model, the M-step has closed form analytical solution. The parameter $\theta$ for the edge weight can be updated with the following relationship,

$$\hat{\theta}(\underline{w}_{ij}) = \log\left[\hat{\eta}(\underline{w}_{ij})\right] + \psi(\hat{\boldsymbol{\theta}}). \tag{14}$$

Similarly, the parameters for the hidden nodes can be updated with the following relationship,

$$\hat{\theta}\left(h_k^{(t)}\right) = \log\left[\hat{\eta}\left(h_k^{(t)}\right) - \sum_i \hat{p}\left(\underline{w}_{ik}^{(t-1)}; \hat{\theta}\right) - \sum_j \hat{p}\left(\underline{w}_{kj}^{(t)}; \hat{\theta}\right)\right] + \psi(\hat{\boldsymbol{\theta}}). \tag{15}$$

We repeat the E- and M-step until the model has converged.

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

## A  Example of Classical Neural Network Architecture

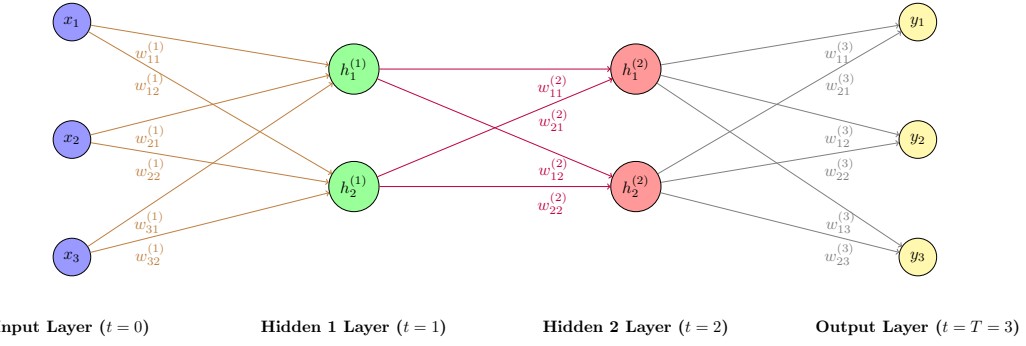

Figure 2: Illustration of a neural network with 3 nodes in the input layer and output layer and 2 hidden layers each with 2 hidden nodes

## B Derivations

### B.1 Optimization: Minimizing KL Divergence

$$D_{\mathrm{KL}}\left(p\left(\mathbf{x},\mathbf{y}\right)\|\hat{p}\left(\mathcal{X},\mathcal{Y},\mathcal{H},\mathcal{W};\boldsymbol{\theta}\right)\right) \tag{16}$$

$$=-\sum_{i=1}^{N}p\left(x_i,y_i\right)\log\frac{\hat{p}\left(\mathcal{H},\mathcal{W},\mathcal{X},\mathcal{Y};\boldsymbol{\theta}\right)}{p\left(x_i,y_i\right)} \tag{17}$$

$$=-\sum_{i=1}^{N}p\left(x_i,y_i\right)\log\frac{\hat{p}\left(\mathcal{H},\mathcal{W}|\mathcal{X},\mathcal{Y};\boldsymbol{\theta}\right)\hat{p}\left(\mathcal{X},\mathcal{Y}\right)}{p\left(x_i,y_i\right)} \tag{18}$$

$$=-\sum_{i=1}^{N}p\left(x_i,y_i\right)\log\hat{p}\left(\mathcal{H},\mathcal{W}|\mathcal{X},\mathcal{Y};\boldsymbol{\theta}\right)-\sum_{i=1}^{N}p\left(x_i,y_i\right)\log\frac{\hat{p}\left(\mathcal{X},\mathcal{Y}\right)}{p\left(x_i,y_i\right)} \tag{19}$$

$$=-\mathbb{E}_{p(\mathbf{x},\mathbf{y})}\left[\log\hat{p}\left(\mathcal{H},\mathcal{W}|\mathcal{X},\mathcal{Y};\boldsymbol{\theta}\right)\right]+D_{\mathrm{KL}}\left(p\left(\mathbf{x},\mathbf{y}\right)\|\hat{p}\left(\mathcal{X},\mathcal{Y}\right)\right) \tag{20}$$

### B.2 Update for Input and Ouput Nodes

$$\hat{\eta}\left(\underline{x}_k\right)=\hat{p}\left(\underline{x}_k\right)+\sum_{j}\hat{p}\left(\underline{w}_{kj};\hat{\theta}\right) \tag{21}$$

$$=\frac{1}{\exp\left[\psi\left(\hat{\boldsymbol{\theta}}\right)\right]}\exp\left[\sum_{s\in\mathcal{S}}\mathbf{1}_{[s\preceq\underline{x}]}\hat{\theta}\left(s\right)\right]+\sum_{j}\frac{1}{\exp\left[\psi\left(\hat{\boldsymbol{\theta}}\right)\right]}\exp\left[\sum_{s\in\mathcal{S}}\mathbf{1}_{[s\preceq\underline{w}_{kj}]}\hat{\theta}\left(s\right)\right] \tag{22}$$

$$=\frac{1}{\exp\left[\psi\left(\hat{\boldsymbol{\theta}}\right)\right]}\exp\left[\hat{\theta}\left(\underline{x}_k\right)\right]+\sum_{j}\frac{1}{\exp\left[\psi\left(\hat{\boldsymbol{\theta}}\right)\right]}\exp\left[\hat{\theta}\left(\underline{x}_k\right)+\hat{\theta}\left(\underline{w}_{kj}\right)+\hat{\theta}\left(\underline{h}_j\right)\right] \tag{23}$$

$$=\frac{1}{\exp\left[\psi\left(\hat{\boldsymbol{\theta}}\right)\right]}\exp\left[\hat{\theta}\left(\underline{x}_k\right)\right]\left[1+\sum_{j}\exp\left[\hat{\theta}\left(\underline{w}_{kj}\right)+\hat{\theta}\left(\underline{h}_j\right)\right]\right] \tag{24}$$

$$=\hat{p}\left(\underline{x}_k\right)\left[1+\sum_{j}\exp\left[\hat{\theta}\left(\underline{w}_{kj}\right)+\hat{\theta}\left(\underline{h}_j\right)\right]\right] \tag{25}$$

$$\hat{p}\left(\underline{x}_k\right)=\frac{\hat{\eta}\left(\underline{x}_k\right)}{\left[1+\sum_{j}\exp\left[\hat{\theta}\left(\underline{w}_{kj}\right)+\hat{\theta}\left(\underline{h}_j\right)\right]\right]} \tag{26}$$

Similarly for $\underline{y}_k$,

$$\hat{p}\left(\underline{y}_k\right)=\frac{\hat{\eta}\left(\underline{y}_k\right)}{\left[1+\sum_{j}\exp\left[\hat{\theta}\left(\underline{w}_{jk}\right)+\hat{\theta}\left(\underline{h}_j\right)\right]\right]} \tag{27}$$

### B.3 Forward Propagation

From Equation (1), we substitute the equivalent values for our partial order formulation.

$$u_k^{(t)}=\epsilon\left(\underline{h}_k^{(t)};\hat{\theta}\right)=\sum_{i=0}^{n}w_{ik}\sigma\left[u_i^{(t-1)}\right]=\sum_{i}\epsilon\left(\underline{w}_{ik}^{(t-1)};\hat{\theta}\right)\sigma\left[\epsilon\left(\underline{n}_i^{(t-1)};\hat{\theta}\right)\right], \tag{28}$$

$$=\sum_{i}\left[\left(\sum_{\omega\in\Omega}\mathbf{1}_{[\omega\preceq\underline{w}_{ik}^{(t-1)}]}\hat{\theta}\left(\omega\right)\right)\cdot\sigma\left(\sum_{\omega\in\Omega}\mathbf{1}_{[\omega\preceq\underline{n}_i^{(t-1)}]}\hat{\theta}\left(\omega\right)\right)\right], \tag{29}$$

$$=\sum_{i}\left[\left(\hat{\theta}\left(\underline{n}_i^{(t-1)}\right)+\hat{\theta}\left(\underline{w}_{ik}^{(t-1)}\right)+\hat{\theta}\left(\underline{n}_k^{(t)}\right)\right)\right]\sigma\left[\hat{\theta}\left(\underline{n}_i^{(t-1)}\right)\right]. \tag{30}$$

The equation can be vectorized for more efficient implementation. The vectorized form is given as,

$$u_k^{(t)} = \left[\hat{\boldsymbol{\theta}}\left(\underline{\mathbf{n}}^{(t-1)}\right) + \hat{\boldsymbol{\theta}}\left(\underline{\mathbf{w}}_{.k}^{(t-1)}\right)\right]\sigma\left[\hat{\boldsymbol{\theta}}^T\left(\underline{\mathbf{n}}^{(t-1)}\right)\right] + \hat{\theta}\left(\underline{n}_k^{(t)}\right)\sum_i \sigma\left[\hat{\theta}\left(\underline{n}_i^{(t-1)}\right)\right], \quad (31)$$

where the boldfaced symbol denotes vectorization across the first dimension of the edge weight index.

## C  Discussion on a Universal Approximator

We have drawn parallels between our approach proposed approach and the neural network. We now provide the connection that our partial order structure is a universal approximator by using the *Kolmogorov-Arnold representation theorem* which states as follows:

**Theorem 1** (Kolmogorov–Arnold Representation Theorem (Braun and Griebel, 2009; Kolmogorov, 1957))**.** *Any multivariate continuous function can be represented as a superposition of one–dimensional functions, i.e.,* $u_k^{(t)} = f\left(u_1^{(t-1)}, \ldots, u_{M^{(t-1)}}^{(t-1)}\right) = \sum_{l=1}^{2M^{(t-1)}+1} f_l\left(\sum_{m=1}^{M^{(t-1)}} g_{l,m}\left(u_m^{(t-1)}\right)\right).$

Braun (2009) showed that the Generalized Additive Model (GAM) is an approximation to the general form presented in Kolmogorov-Arnold representation theorem by replacing the range $l \in \{1, \ldots, 2M^{(t-1)} + 1\}$ with our partial order structure and the inner function $g_{l,m}$ by the identity if $m = l$ and zero otherwise, yielding $u_k^{(t)} = f(u_1^{(t-1)}, \ldots, u_{M^{(t-1)}}^{(t-1)}) = \sum_{m=1}^{M^{(t-1)}} f_m(u_m^{(t-1)})$ , where $f_m(\cdot)$ is a smooth monotonic function. For our model this is the function for forward propagation in Equation (12) which is a smooth monotonic function with respect to the input $u_m^{(t-1)}$. The model is able to approximate the input $\hat{o}$ to be $|\hat{o} - \sigma(u_k^{(t)})| < \delta$, where $\delta$ is the error in the approximation. By applying the forward propagation function through different layers, we can generalize the expression to be $u_k^{(t)} = f(u_1^{(\tau)}, \ldots, u_{M^{(\tau)}}^{(\tau)}) = \sum_{m=1}^{M^{(\tau)}} f_m^{(t-1)} \circ f_m^{(t-2)} \circ \ldots \circ f_m^{(\tau+1)} \circ f_m^{(\tau)}(u_m^{(\tau)}) = \sum_{m=1}^{M^{(\tau)}} F_m^{(t,\tau)}(u_m^{(\tau)})$, where $\tau$ represents a layer in the neural network that is $\tau < t$. The function $F_m^{(t,\tau)}$ remains a smooth monotonic function because when we stack multiple smooth monotonic functions together, the overall transformation by the function is still smooth and monotonic. This means for any giving input $\mathbf{x}$ we are always able to learn a representation for the given output $\mathbf{y}$.

## D  Summary

We have presented a novel perspective on deep learning architectures using a partial order structure which can be naturally represented as a log-linear model studied in information geometry. We first have shown that a partial order structure can be used to represent deep architectures. We then formulated our optimization is formulated by minimizing the KL-divergence between the set of inputs and our partial order structure. We use the EM algorithm for optimization and it has a closed form analytical solution for both the E- and the M-step and does not require any gradients for optimization. Our approach has clear advantages as it does not have the same drawbacks in the classical deep learning models such as vanishing gradients.

