# OpenReview forum: "A Deep Architecture for Log-Linear Models"
_NeurIPS.cc/2020/Workshop/DL-IG — NeurIPSW 2020: DL-IG Poster_

### Official Review · AnonReviewer1 · 2020-11-01

**Rating:** 7
**Confidence:** 4

**Review:**

This paper investigates a partial-order on the variables in a neural network. The authors, motivated from log-linear models and a poset structure on its variables, construct a similar model for a deep neural network. The parameters of the model are optimized using the expectation-maximization (EM) algorithm. The paper derives the expressions for these updates. This approach is interesting and potentially beneficial because the EM steps do not require computing back-propagation gradients, both steps can be executed using closed-form formulae.

The paper will benefit from an empirical study, where this method can perhaps be compared to a Boltzmann machine.

---

### Author Response · Authors · 2020-12-11
**NeurIPS2020 Workshop Content**

The link to the 5 minute presentation is:
https://youtu.be/B1F5Ujtb8zQ

The link to the poster is:
https://github.com/sjmluo/PosetDL/blob/master/PosetDL_NeurIPS2020_Poster.pdf

---

### Decision · Program_Chairs · 2020-11-07

Accept (Poster)